# Pachydrusen in Fellow Eyes Predict Response to Aflibercept Monotherapy in Patients with Polypoidal Choroidal Vasculopathy

**DOI:** 10.3390/jcm9082459

**Published:** 2020-07-31

**Authors:** Yoshiko Fukuda, Yoichi Sakurada, Atsushi Sugiyama, Seigo Yoneyama, Mio Matsubara, Wataru Kikushima, Naohiko Tanabe, Ravi Parikh, Kenji Kashiwagi

**Affiliations:** 1Departments of Ophthalmology, Faculty of Medicine, University of Yamanashi, Chuo 409-3898, Yamanashi, Japan; ysugiyama@yamanashi.ac.jp (Y.F.); asugiyama@yamaanshi.ac.jp (A.S.); syoneyama@yamanashi.ac.jp (S.Y.); miom@yamanashi.ac.jp (M.M.); wkikushima@yamanashi.ac.jp (W.K.); tanabe@yamanashi.ac.jp (N.T.); kenjik@yamanashi.ac.jp (K.K.); 2Department of Ophthalmology, School of Medicine, New York University, New York, NY 10016, USA; parikh815@gmail.com; 3Manhattan Retina and Eye Consultants, New York, NY 10016, USA

**Keywords:** as-needed aflibercept therapy, polypoidal choroidal vasculopathy, pachydrusen, treatment burden

## Abstract

We investigated whether responses to as-needed intravitreal aflibercept injections (IAIs) for polypoidal choroidal vasculopathy (PCV) differed among patients based upon drusen characteristics in fellow eyes. 110 eyes from 110 patients with PCV received 3 monthly IAI and thereafter Pro re nata (PRN) IAI over 12 months. Patients were classified into 4 groups depending on fellow eye findings. Group 1 (n = 16): pachydrusen; Group 2 (n = 45): no drusen; Group 3 (n = 35): soft drusen; Group4 (n = 14) PCV/scarring. Best-corrected visual acuity improved at 12 months in all groups, but not significantly in Group 1 and Group 4; however, visual improvement was similar among the groups after adjusting baseline confounders. Group 1 had a significantly lower percentage of eyes needing retreatment (all *p* < 0.001; Group 1: 16.7%; Group 2: 50.8%; Group 3: 80%; Group 4: 85.7%). The mean number of retreatments was least in Group 1 among the groups (all *p*-value < 0.003; Group 1: 0.50 ± 1.32; Group 2: 1.73 ± 2.08; Group 3:2.71 ± 1.99; Group 3: 2.71 ± 2.16). Patients with pachydrusen in fellow eyes were less likely to require additional IAI following the loading dose and may be ideal candidates for aflibercept monotherapy in their first year.

## 1. Introduction

Polypoidal choroidal vasculopathy (PCV) is considered a variant of exudative age-related macular degeneration (AMD) characterized by clinical features including type 1 neovascularization (located between retinal pigment epithelium and Bruch’s membrane) with terminal aneurysmal dilation, seen on indocyanine green angiography (ICGA) [1]. In a Japanese clinic-based study, the prevalence of PCV was reported to be almost half of the patients with advanced AMD [2].

Pachydrusen are clinical entities characterized by isolated and scattered yellowish drusenoid deposits over 125 µm in diameter and in areas of increased choroidal thickness [3,4].

In 2005, genome wide association studies revealed that *Complement factor H* (*CFH)* Y402H and *Age-related maculopathy susceptibility2* (*ARMS2)* A69S were strongly associated with AMD in Caucasians [5,6]. Subsequent studies confirmed that these genes were associated with neovascular AMD as well as PCV and retinal angiomatous proliferation [7,8]. Unlike Caucasians, I62V at *CFH* gene was found to have the strongest association with AMD in Asians [9].

It has been reported that patients with pachydrusen are similar to patients without drusen in terms of risk allele frequency of *ARMS2* and *CFH*, suggesting that pachydrusen are not risk of exudative AMD [10].

To date, intravitreal injections of vascular endothelial growth (VEGF) inhibitors have become standard treatment for PCV along with photodynamic therapy combined with anti-VEGF agents [11,12,13,14]. Aflibercept is a recombinant fusion protein that binds to VEGF-A, VEGF-B, and placental growth factor. It has been reported that its binding affinity is higher compared with ranibizumab and bevacizumab [15]. Several studies reported that intravitreal aflibercept therapy is effective for eyes with PCV in terms of improving vision and morphology with different treatment regimens [16,17].

A previous study reported that untreated fellow eye condition might be a factor predictive for treatment response to VEGF inhibitors therapy for neovascular AMD [18].

In the present study, we assessed the untreated fellow eye condition in patients with PCV who received intravitreal aflibercept injection and classified the fellow eyes into 4 groups; Group 1: pachydrusen; Group 2: no drusen; Group 3: soft drusen; Group 4: PCV/Scarring and investigated whether response to intravitreal aflibercept therapy for Group 1 was different from those for other groups.

## 2. Methods

### 2.1. Subjects

A retrospective medical chart was reviewed for consecutive 110 patients with treatment naïve PCV receiving 3 monthly intravitreal aflibercept injections (IAIs) at Macula Clinic of University of Yamanashi between January 2013 and January 2019. The inclusion criteria were 1) symptomatic PCV with foveal exudation 2) follow-up period equal or more than 12 months 3) baseline best-corrected visual acuity from 0.06 to 1.0 in decimal format. The exclusion criteria were 1) other exudative maculopathy including typical neovascular AMD, retinal angiomatous proliferation, angioid streaks, and myopic neovascularization 2) opacity media such that fundus cannot be seen at least one eye, 3) previous treatment history for PCV including photodynamic therapy, intravitreal injection of other anti-VEGF agents such as ranibizumab or bevacizumab.

PCV was diagnosed as we previously described [2]. All PCV cases exhibited solitary/ multiple aneurysmal bulges (polypoidal lesions) with or without branching vascular networks as seen on ICGA and retinal pigment epithelium (RPE) protrusion with serous/hemorrhagic detachment of the neurosensory retina and/or RPE on spectral-domain optical coherence tomography (SD-OCT).

The approval for the retrospective study was acquired by the institutional review board at the University of Yamanashi (Research No.2149) and followed the tenets of Declaration of Helsinki. Prior to IAI, written informed consent for treatment was obtained from each patient.

### 2.2. Group Classification

Based on color fundus photograph, spectral-domain optical coherence tomography (SD-OCT) using Spectralis version 5.4 HRA + OCT (Heidelberg Engineering, Dossenheim, Germany) and fluorescein angiography(FA) and indocyanine green angiography(ICGA) using a confocal laser scanning system (HRA-2;Heidelberg Engineering, Dossenheim, Germany), we classified patients into 4 groups depending on untreated fellow eye condition; Group 1: pachydrusen; Group 2: no drusen; Group 3: soft drusen; Group 4: PCV/Scarring. Soft drusen exhibited hyperreflective RPE elevation SD-OCT and hypofluorescent on late phase ICGA (Figure 1) and the size of drusen greater than 63 µm was defined as presence of soft drusen on the color fundus photography. On the other hand, pachydrusen showed drusenoid deposit on SD-OCT and hyperfluorescent on late phase ICGA (Figure 2) and the size of drusenoid deposit greater than 125 µm was defined as presence of pachydrusen on the color fundus photography. Two independent graders (Y.F and Y.S) judged the groups. Discordant gradings were resolved through open discussion.

### 2.3. Treatment and Follow-Up

Prior to the treatment, all patients received a comprehensive examination including best-corrected visual acuity (BCVA) measurement using Landolt chart, biomicroscopy with 78 D lens, spectral-domain optical coherence tomography (SD-OCT) using Spectralis version 5.4 (Heidelberg Engineering, Dossenheim, Germany), fluorescein/indocyanine green angiography (FA/ICGA) was spontaneously performed using Spectralis. All eyes with PCV demonstrated characteristic finding with solitary/multiple polypoidal dilations with or without branching vascular network as seen on ICGA. Most aneurysmal dilations on ICGA corresponded to orange-red lesion in color fundus photography.

Three monthly IAIs (2.0mg/0.05mL) was administrated for all patients and thereafter patients were followed monthly with SD-OCT imaging and a dilated fundus exam. If exudation were seen on SD-OCT or new hemorrhage beneath the retina or RPE was seen on ophthalmic fundus examination, additional IAI was performed.

The greatest linear dimension, which was defined as the lesion including polypoidal lesion (hot spot lesions) and branching vascular network, was determined on ICGA images. Subfoveal choroidal thickness was measured for all patients at baseline, 3 months, 6 months, 9 months, 12 months and defined as the vertical distance between choroidoscleral border and outer border of RPE by means of SD-OCT. Central retinal thickness was also measured for all patients at baseline, 3 months, 6 months, 9 months, 12 months and defined as the vertical distance between inner border of RPE and inner surface of neurosensory retina at the fovea by SD-OCT.

### 2.4. Genotyping

Age-related maculopathy susceptible2 (*ARMS2*) A69S (rs10490924) and Complement factor H (*CFH*) I62V (rs800292) was performed for all participants. A peripheral blood was collected from each participant when FA/ICGA was performed. A purified DNA was obtained by a Pure Gene DNA Isolation Kit (Gentra Systems, Minneapolis, Minnesota, USA). Genotyping was conducted using TaqMan genotyping assays with a 7300/7500 Real-Time PCR System (Applied Biosystems, CA, USA) in accordance with the manufacturer’s introduction as we previously described [19].

## 3. Statistical Analysis

Statistical analyses were conducted using DR SPSS (IBM, Tokyo, Japan). BCVA measured in the decimal scale with Landolt chart was converted into a logarithm of the minimal angle resolution (log MAR) unit for statistical use. The differences of categorical variables between the 2 groups were evaluated by chi-square test. The differences of continuous variables were evaluated by Mann-Whitney U test or Kruskal-Wallis test. A Kaplan–Meier estimator showing the retreatment-free proportion was constructed and a log-rank test was used to compare a difference of cumulative retreatment-free proportion between 2 groups. A paired t-test was used to compare the values between pretreatment and posttreatment. A *p*-value of less than 0.05 were considered a statistical significance.

## 4. Results

Baseline demographic and clinical characteristics of patients with PCV is shown in Table 1. Group 1 was youngest among the 4 groups. (*p* = 3.4×10^-4^) and subfoveal choroidal thickness in Group 1 was greatest among the 4 groups. (*p* = 2.2×10^-3^) T allele frequency was significantly lower in Group 1 compared with other groups. (all *p*-values < 0.05) Pseudodrusen were not seen in this cohort.

BCVA improved in all groups at 12 months: although BCVA significantly improved at 12 month in Group 2 (*p* = 4.8×10^-7^), a significant improvement was not seen in Group 1 (*p* = 0.08); Group 3 (*p* = 0.14), and Group 4 (*p* = 0.054), Figure 3. Change of central retinal thickness, subfoveal choroidal thickness are in Figure 4 and Figure 5. Figure 6 shows Kaplan-Meier estimator regrading retreatment-free proportion from initial treatment. However, visual improvement was not significantly different between Group 1 and other groups after adjusting for baseline confounders including age, gender, BCVA, greatest linear dimension, central retinal thickness, and subfoveal choroidal thickness.(vs Group 2, *p* = 0.38, vs Group 3, *p* = 0.26, vs Group 4, *p* = 0.97). Change of central retinal thickness, subfoveal choroidal thickness are in Figure 4 and Figure 5. Figure 6 shows Kaplan-Meier estimator regrading retreatment-free proportion from initial treatment.

## 5. Discussion

In the present study, we assessed how fellow eye drusen characteristics predicted response to IAI in terms of BCVA and need for retreatment. Although there was not a significant difference in visual improvement among the 4 groups, patients with pachydrusen in fellow eyes were less likely to recur, requiring fewer additional injections compared to other groups beyond the initial 3 monthly IAI.

There are a few possible explanations for our present outcomes. First, several studies demonstrated that older age was associated with retreatment when treating with anti-VEGF monotherapy including ranibizumab or aflibercept [20,21]. In the present study, patients with pachydrusen in their fellow eye (Group 1) were younger compared to the other groups. Second, risk allele (T allele) frequency of *ARMS2* A69S was significantly lower in Group 1 patients compared to those in other groups, which has previously reported [10]. It is well known that *ARMS2* A69S and *CFH* I62V are major genetic variants susceptible to exudative AMD including PCV in Asians [9,22]. Moreover, it has been reported that risk variants of *ARMS2* A69S is associated with severe phenotype and bilateral involvement and that protective variants of *ARMS2* A69S are associated with subfoveal choroidal thickness [23,24,25,26]. Previous studies demonstrated that those with risk allele homozygosity for *ARMS2* A69S, but not *CFH* I62V, were likely to recur and require more intravitreal injections compared to other genotypes when treating with PRN anti-VEGF therapy [27,28,29,30]. Third, pachydrusen were considered pachychoroid-driven products, but not inflammatory products of complement system as with soft drusen. Indeed, it has been reported that aqueous humor inflammatory cytokines levels were lower in pachychoroid neovasculopathy compared to neovascular AMD [31]. Even if achieving dry macula after the loading phase, recurrent exudation might be more likely to occur in drusen-driven AMD compared with pachychoroid-driven AMD.

In the present study, we demonstrated that drusen characteristics in fellow eyes were clinical predictors of IAI retreatment, and retreatment-free period, without genotyping *ARMS2* A69S variants. However, it is sometime difficult to differentiate scattered soft drusen from pachydrusen because appearance of two entities are similar on color fundus photography. In the present study, we differentiated two entities using multimodal imaging including SD-OCT and late phase ICGA. Drusen contain a variety of components such as lipid and inflammatory components [32]. The risk of advanced AMD differs depending on drusen size, type/subtypes and fellow eye condition as well as retinal pigment epithelial changes [33,34,35]. Depending on drusen type, inflammatory reactions in the retina might be also different. Therefore, it is plausible that drusen type in fellow eyes of PCV patients may have some bearing on IAI response; however, further research is needed.

Limitations of the present study are its retrospective nature of analysis and small sample size. To confirm or refute this tentative conclusion, a large-scale study would be needed.

In summary, PCV patients with pachydrusen in fellow eyes were less likely to require additional IAI following the loading dose of 3 monthly IAI and may be ideal candidates for aflibercept monotherapy in their first year. Among PCV patients, fellow eye pachydrusen could be a predictive biomarker for a benign course indicating a decreased probability of recurrence and treatment burden.

## Figures and Tables

**Figure 1 jcm-09-02459-f001:**
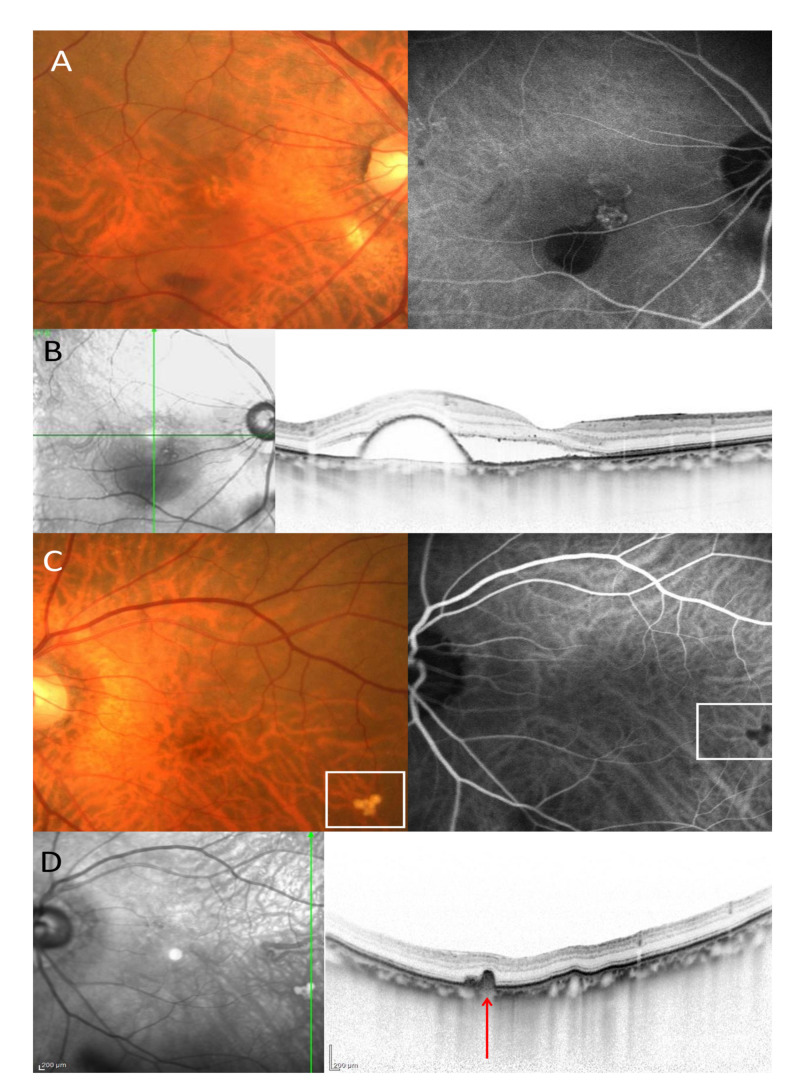
An eighty-year-old male patients with polypoidal choroidal vasculopathy. (**A**). Color fundus photography showed hemorrhagic pigment epithelial detachment in the right eye. On the indocyanine green angiography (ICGA), polypoidal lesion and branching vascular network was seen in the right eye. (**B**). A vertical OCT scan through the fovea showed subretinal fluid and pigment epithelial detachment. (**C**). Color fundus photography showed soft drusen surrounded by a white square in the left eye. On the ICGA, the soft drusen exhibited hypofluorescent in the left eye. (**D**). A vertical OCT scan through the soft drusen showed retinal pigment epithelial bump (red arrow).

**Figure 2 jcm-09-02459-f002:**
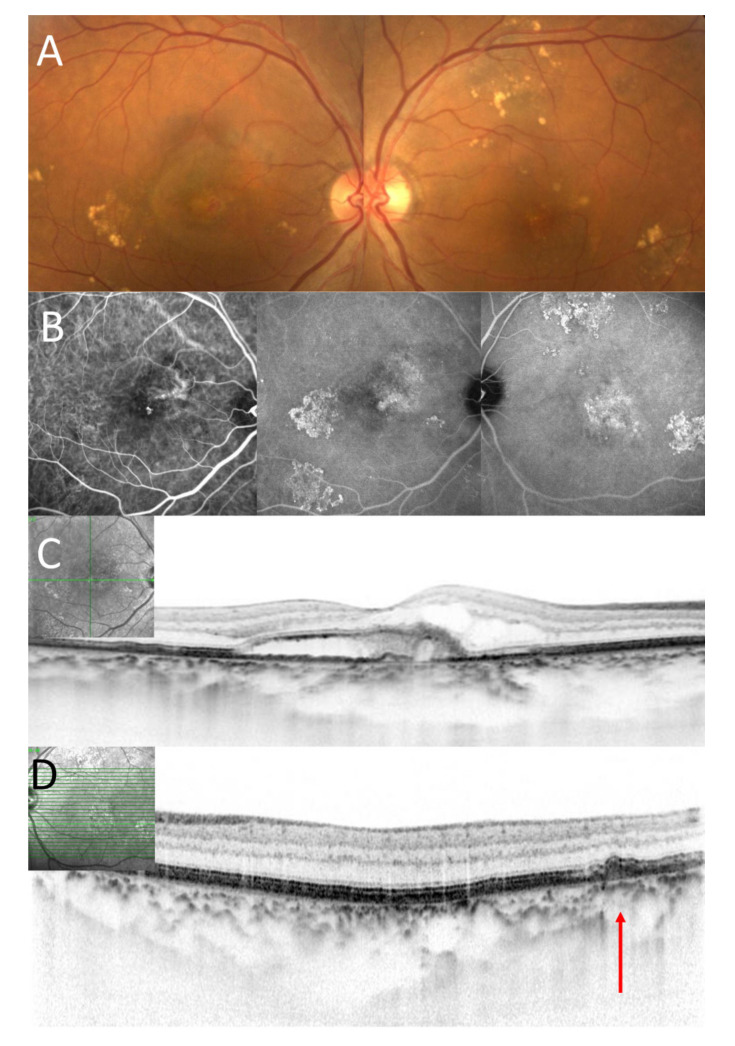
A seventy-three-year-old male patient with polypoidal choroidal vasculopathy. (**A**). Color fundus photography showed scattered yellowish drusenoid deposits around the macular in both eyes and exudation in the macular in the right eye. (**B**). On the early phase of ICGA, polypoidal lesion was seen in the macula in the right eye. (Left image). On the late phase of ICGA, hyperfluorescence was seen in the area corresponding to scattered yellowish drusenoid deposits in both eyes. (Middle and left image) (**C**). A horizontal OCT scan through the fovea showed exudation including subretinal fluid and fibrin in the right eye. Subfoveal choroidal thickness was 375 µm in the right eye. (**D**). A horizontal OCT scan through the yellowish deposits showed drusenoid deposits (as indicated by a red arrow) in the left eye.

**Figure 3 jcm-09-02459-f003:**
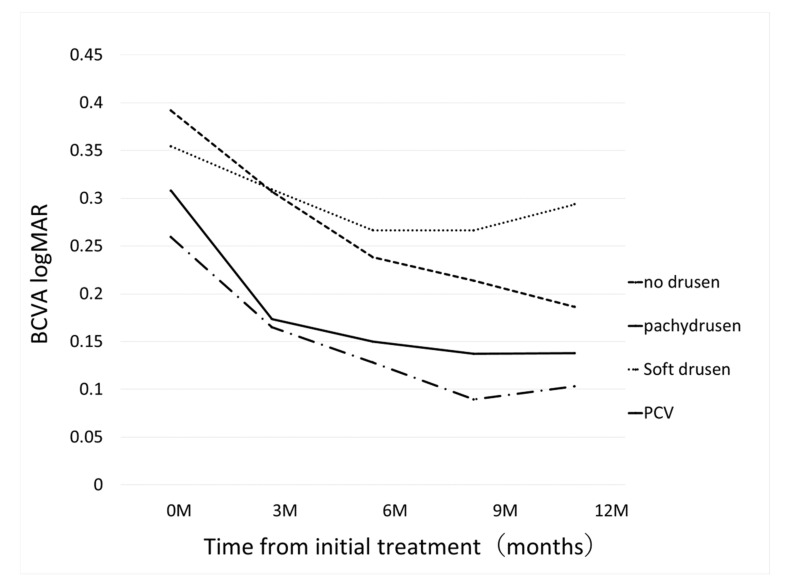
Changes of best-corrected visual acuity (BCVA) from initial treatment according to the groups. In pachydrusen group, BCVA improved from 0.26 ± 0.32 (baseline) to 0.17 ± 0.19 (*p* = 0.276), 0.13 ± 0.20 (p = 0.11), 0.089 ± 0.16 (*p* = 0.054), 0.10 ± 0.20 (*p* = 0.081) at 3 months, 6 months, 9 months, and 12 months, respectively. In no drusen group, BCVA improved from 0.39 ± 0.38 (baseline) to 0.31 ± 0.34 (*p* = 0.023), 0.24 ± 0.32 (*p* = 3.8×10^-6^), 0.21 ± 0.29 (*p* = 8.25×10^-8^), 0.19 ± 0.28 (*p* = 4.8×10^-7^) at 3 months, 6 months, 9 months, and 12 months, respectively. In soft drusen group, BCVA improved from 0.35 ± 0.29 (baseline) to 0.31 ± 0.35 (*p* = 0.016), 0.27 ± 0.31 (p = 9.8×10^-3^), 0.27 ± 0.34 (*p* = 0.043), 0.29 ± 0.35 (*p* = 0.14) at 3 months, 6 months, 9 months, and 12 months, respectively. In PCV/Scarring group, BCVA improved from 0.31 ± 0.35 (baseline) to 0.17 ± 0.19 (*p* = 0.086), 0.15 ± 0.20 (*p* = 0.019), 0.14 ± 0.16 (*p* = 0.074), 0.14 ± 0.14 (*p* = 0.054) at 3 months, 6 months, 9 months, and 12 months, respectively.

**Figure 4 jcm-09-02459-f004:**
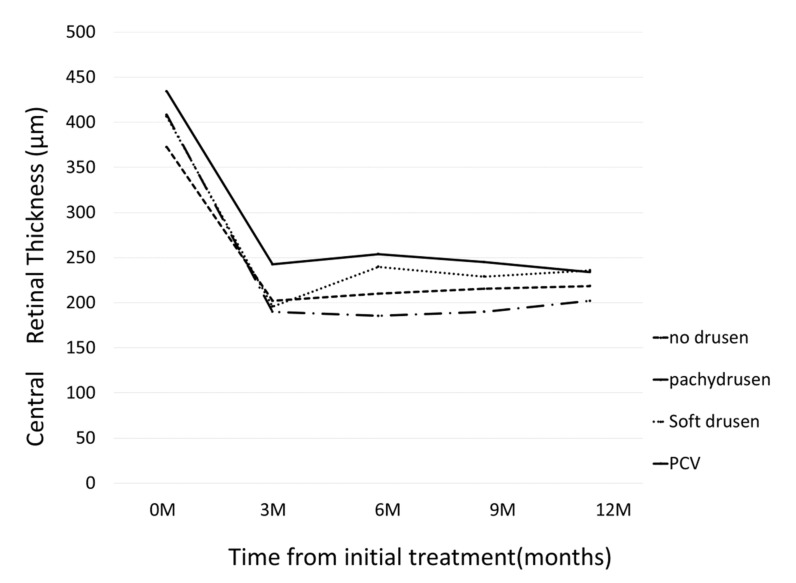
Changes of central retinal thickness (CRT) from initial treatment according to the groups. In pachydrusen group, CRT significantly decreased from 408 ± 171 (baseline) to 190 ± 98, 185 ± 82, 190 ± 86, 202 ± 113 at 3 months, 6 months, 9 months, and 12 months, respectively. (all *p*-value < 0.001) In no drusen group, CRT significantly decreased from 373 ± 160 (baseline) to 202 ± 81, 210 ± 71, 215 ± 75, 218 ± 85 at 3 months, 6 months, 9 months, and 12 months, respectively. (all *p*-value < 0.001) In soft drusen group, CRT significantly decreased from 406 ± 210 (baseline) to 196 ± 69, 240 ± 113, 229 ± 114, 236 ± 122 at 3 months, 6 months, 9 months, and 12 months, respectively. (all *p*-value < 0.001) In PCV/Scarring group, CRT significantly decreased from 434 ± 132 (baseline) to 242 ± 99, 254 ± 114, 245.0 ± 84, 234 ± 83 at 3 months, 6 months, 9 months, and 12 months, respectively. (all *p*-value < 0.05).

**Figure 5 jcm-09-02459-f005:**
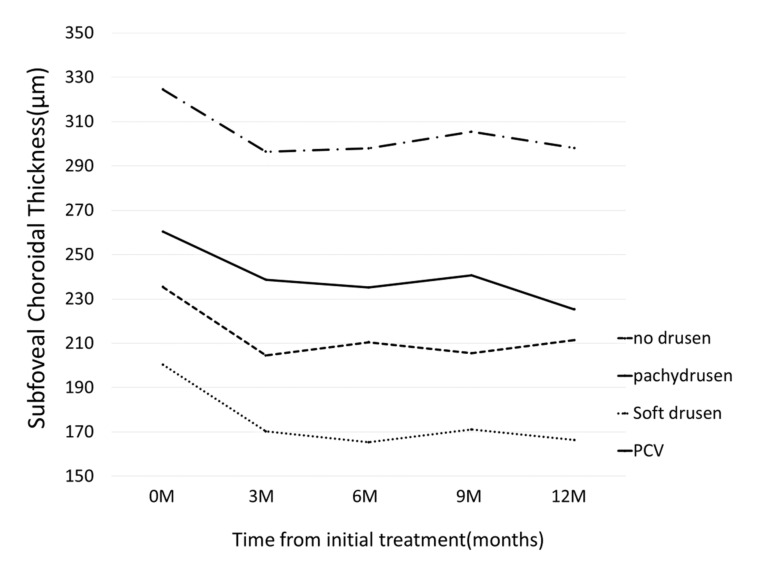
Changes of subfoveal choroidal thickness (SCT) from initial treatment according to the groups. In pachydrusen group, SCT significantly decreased from 325 ± 114 (baseline) to 296 ± 108, 298 ± 120, 305 ± 119, 298 ± 118 at 3 months, 6 months, 9 months, and 12 months, respectively. (all *p*-value < 0.05) In no drusen group, SCT significantly decreased from 235 ± 92 (baseline) to 204 ± 82, 210 ± 89, 206 ± 83, 211 ± 88 at 3 months, 6 months, 9 months, and 12 months, respectively. (all *p*-value < 0.0001) In soft drusen group, SCT significantly decreased from 200 ± 76 (baseline) to 170 ± 77, 165 ± 72, 171 ± 72, 166 ± 67 at 3 months, 6 months, 9 months, and 12 months, respectively. (all *p*-value < 0.01) In PCV/Scarring group, SCT significantly decreased from 260 ± 114 (baseline) to 239 ± 100, 235 ± 93, 241 ± 97, 225 ± 89 at 3 months, 6 months, 9 months, and 12 months, respectively. (all *p*-value < 0.05).

**Figure 6 jcm-09-02459-f006:**
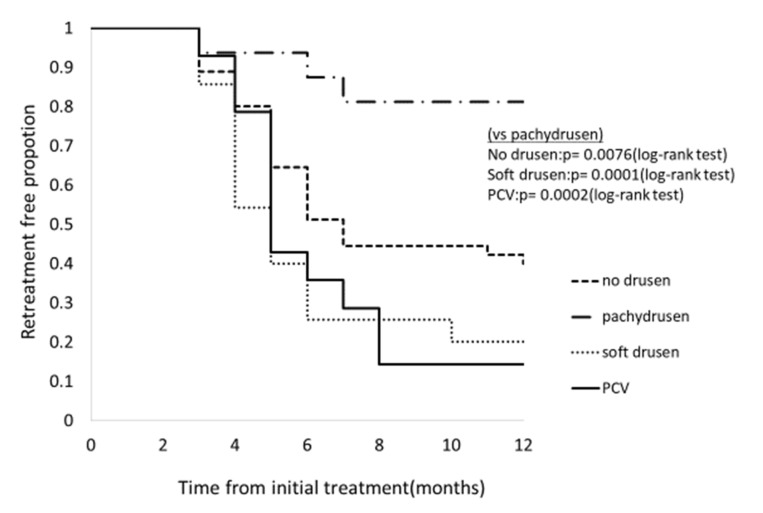
Kaplan–Meier estimator showing retreatment-free proportion after initial treatment according to the groups. Retreatment-free proportion at 12 months was 83.3% (13/16), 41.2% (19/45), 20% (7/35), 14.3% (2/14) in pachydrusen group, no drusen group, and PCV/ Scarring group, respectively. Retreatment-free period was significantly longer in pachydrusen group compared to other groups. (all *p*-value < 0.001). Mean number of additional injections was significantly smaller in Group 1 (0.5 ± 1.32) compared with other groups. (Group 2: 1.73 ± 2.08, *p* = 7.5×10^-3^; Group 3: 2.71 ± 1.99, *p* = 1.4×10^-4^; Group 4: 2.71 ± 2.16, *p* = 4.0×10^-4^).

**Table 1 jcm-09-02459-t001:** Baseline demographic and clinical characteristics of patients with polypoidal choroidal vasculopathy.

	Group 1: Pachydrusen (n = 16)	Group 2: No Drusen (n = 45)	Group 3: Soft Drusen (n = 35)	Group 4: PCV/Scarring (n = 14)	*p*-value
Mean age(year)	68.4	69.7	75.0	77.9	3.4×10^-4^☨
*p*-value (vs Pachydrusen)	NA	0.55☨☨	3.1×10^-3^☨☨	2.5×10^-3^☨☨	
Male (%)	12 (75%)	37 (73.3%)	27 (77.14%)	10 (71.43%)	0.97☨☨☨
p-value (vs Pachydrusen)	NA	0.90☨☨☨	0.87☨☨☨	0.83☨☨☨	
BCVA log MAR	0.26 ± 0.32	0.39 ± 0.38	0.35 ± 0.29	0.31 ± 0.35	0.45☨
p-value (vs Pachydrusen)	NA	0.18☨☨	0.15☨☨	0.75☨☨	
Mean central retinal thickness(µm)	408	373	406	434	0.26☨
p-value (vs Pachydrusen)	NA	0.56☨☨	0.66☨☨	0.39☨☨	
Mean subfoveal choroidal thickness(µm)	325	235	200	260	2.2×10^-3^☨
p-value (vs Pachydrusen)	NA	6.0×10^-3^☨☨	4.3×10^-3^☨☨	0.21☨☨	
Mean greatest linear dimension(µm)	3771	3669	3464	4761	0.15☨
p-value (vs Pachydrusen)	N/A	0.81☨☨	0.75☨☨	0.16☨☨	
*ARMS2* A69S T allele frequency	0.34	0.57	0.54	0.75	0.020☨☨☨
p-value (vs Pachydrusen)	NA	0.038☨☨☨	0.047☨☨☨	4.0×10^-3^☨☨☨	
TT	1 (6.25%)	16 (35.6%)	9 (25.7%)	9 (64.3%)	
TG	9 (56.3%)	19 (42.2%)	20 (57.1%)	3 (21.4%)	
GG	6 (12.5%)	10 (22.2%)	6 (17.1%)	2 (14.3%)	
*CFH* I62V G allele frequency	0.72	0.74	0.69	0.93	0.054☨☨☨
p-value (vs Pachydrusen)	NA	0.90☨☨☨	0.52☨☨☨	0.070☨☨☨	
GG	9 (56.3%)	25 (55.6%)	14 (40.0%)	12 (85.7%)	
GA	5 (31.3%)	17 (37.8%)	20 (57.1%)	2 (14.3%)	
AA	2 (12.5%)	3 (6.7%)	1 (2.9%)	0 (0%)	

ARMS2: age-related maculopathy susceptible 2; CFH: complement factor H; NA: not applicable; ☨: Kruskal-Wallis test; ☨☨: Mann-Whitney U test; ☨☨☨: chi-square test.

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
