# Peer review of "Pachydrusen in Fellow Eyes Predict Response to Aflibercept Monotherapy in Patients with Polypoidal Choroidal Vasculopathy"

_jcm, 2020, doi:10.3390/jcm9082459_

Round 1
Reviewer 1 Report
In this study, the authors investigated difference in treatment response (especially recurrence needing retreatment) among eyes with different fellow-eye findings. As a result, presence of pachydrusen in the fellow eye was found to be associated with significantly low recurrence rate.
The study is well-designed. But I have several concerns
Results and discussion
The recurrence rate in the pachydrusen group is too low (only 16.7%). Although the authors described potential explanations in the discussion, more effort is needed to more clearly explain this particularly low recurrence rate.
Conclusion
I cannot understand why the result “..less likely to require additional IAI following the loading dose of 3 monthly IAI” lead the conclusion “…may be ideal candidates for aflibercept monotherapy in their first year.” Is there any problem with implementing IAI if the recurrence rate is higher?
Author Response
Reviewer 1
In this study, the authors investigated difference in treatment response (especially recurrence needing retreatment) among eyes with different fellow-eye findings. As a result, presence of pachydrusen in the fellow eye was found to be associated with significantly low recurrence rate.
The study is well-designed. But I have several concerns
Reply: Thank you for positive feedback on our manuscript.
Results and discussion
The recurrence rate in the pachydrusen group is too low (only 16.7%). Although the authors described potential explanations in the discussion, more effort is needed to more clearly explain this particularly low recurrence rate.
Reply: Thank you for a valuable comment. W added a third possible reason in the Discussion. The sentences are as follows. “Third, pachydrusen were considered pachychoroid-driven products, but not inflammatory products of complement system like soft drusen. Indeed, it has been reported that aqueous humor inflammatory cytokines levels were lower in pachychoroid neovasculopathy compared to neovascular AMD.[31] Even if achieving dry macula after the loading phase, recurrent exudation might be more likely to occur in drusen-driven AMD compared to pachychoroid-driven AMD.”
Conclusion
I cannot understand why the result “..less likely to require additional IAI following the loading dose of 3 monthly IAI” lead the conclusion “…may be ideal candidates for aflibercept monotherapy in their first year.” Is there any problem with implementing IAI if the recurrence rate is higher?
Reply: In general, frequent intravitreal injections could be economically and psychologically burdensome for patients.
Reviewer 2 Report
In this study the authors investigated whether the presence of pachydrusen in the fellow eye could predict response to anti vegf treatment of eyes with polypoidal neovasculopathy.
I think it is an interesting study. Could the authors explain whether vegf level in the eyes could be related to a different response to treatment?
Author Response
Reviewer 2
In this study the authors investigated whether the presence of pachydrusen in the fellow eye could predict response to anti vegf treatment of eyes with polypoidal neovasculopathy.
I think it is an interesting study. Could the authors explain whether vegf level in the eyes could be related to a different response to treatment?
Reply: Thank you for your positive feedback. Higher VEGF level in the retina or RPE would be associated with recurrent exudation after 3 monthly IAI. Given that eyes are generally symmetrical, VEGF level in fellow eyes with pachydrusen would be low because pachydrusen appear by pachychoroid-driven mechanism, but not complement system inflammation. On the other hand, VEGF level in fellow eyes with soft drusen are relatively high because soft drusen are products of inflammatory activations.
Reviewer 3 Report
Although the paper is interesting and the methodology is sound, the following concerns need to be addressed before accepting the paper.
Plagiarism check showed a level of 35%, which is quite high. To be considered for publication, authors need to make a thorough revision to keep the Similarity Index ≤ 20%.
Introduction section:
The two SNPs tested in this study, ARMS2 A69S(rs10490924) and CFH I62V(rs800292), were not adequately introduced despite being studied and showing significant results. The authors need to add an adequate rationale explaining their choice of the SNPs. What is the role of the genes ARMS2 and CFH in polypoidal choroidal vasculopathy? Why did they choose these SNPs and not others?
Line 128: As mentioned in the limitation, the authors have conducted their study on limited sample size. Did they try to carry out any sampling/experimental design before conducting the study? Determining the optimal sample size in the statistical analysis (subsection 2.5) study could provide readers about the adequate number of participants needed to detect significant robust results! The GPower tool can do this calculation.
Line 130: The authors have performed Kaplan-Meier estimator analysis (Figure 6) but did not mention it in the statistical analysis subsection. The authors need to mention this.
Results:
Lines 133-134: The authors cite the exact values in the text (68.4 years, 69.7, 75.0, and 77.9) despite being displayed in Table 1. The authors are encouraged to decrease this overlap with the Table and instead summarize these findings. Readers can retrieve this information from the Table.
Line 139: Since one Table only exists, authors do not need to write Table 1. Instead, they can add the Table only.
Line 141: A legend for table 1 is lacking. It is also not enough to put a P-value without mentioning the statistical test that generated this P-value. The authors need to add a legend where they mention what statistical tests were used (chi-squared on genotypes, etc…) and the abbreviation used NA: not applicable.
Minor:
Line 37: Authors must write the acronym for the following genes (ARMS2 and CFH) at their first appearance in the text.
Line 55: add a space between consecutive110. Idem in Line 144. Idem in line 139: Genotyping of ARMS2 A69S(rs10490924) and CFH I62V(rs800292).
The quality of figures 4 and 5 is poor. The authors need to handle this manner appropriately.
Author Response
Reviewer 3
Although the paper is interesting and the methodology is sound, the following concerns need to be addressed before accepting the paper.
Reply: Thank you for timely and constructive feedback on our manuscript.
Plagiarism check showed a level of 35%, which is quite high. To be considered for publication, authors need to make a thorough revision to keep the Similarity Index ≤ 20%.
Reply: Probably the high similarity index is because we used the same description with our previous articles in Methods and Introduction section. We changed the description in Introduction and Methods as possible as we can.
Introduction section:
The two SNPs tested in this study, ARMS2 A69S(rs10490924) and CFH I62V(rs800292), were not adequately introduced despite being studied and showing significant results. The authors need to add an adequate rationale explaining their choice of the SNPs. What is the role of the genes ARMS2 and CFH in polypoidal choroidal vasculopathy? Why did they choose these SNPs and not others?
Reply: We added the following sentences so as that readers can easily understand the history of AMD genetics and the reason to apply both variants of ARMS2 A69S and CFH I62V in the present study. “ In 2005, genome wide association studies revealed that Complement factor H (CFH) Y402H and Age related maculopathy susceptibility2 (ARMS2) A69S were strongly associated with AMD in Caucasians.[5, 6] Subsequent studies confirmed that these genes were associated with neovascular AMD as well as PCV and retinal angiomatous proliferation.[7, 8] Unlike Caucasians, I62V at CFH gene was found to have the strongest association with AMD in Asians.[9]”
Line 128: As mentioned in the limitation, the authors have conducted their study on limited sample size. Did they try to carry out any sampling/experimental design before conducting the study? Determining the optimal sample size in the statistical analysis (subsection 2.5) study could provide readers about the adequate number of participants needed to detect significant robust results! The GPower tool can do this calculation.
Reply: Thank you for a valuable comment. Unfortunately, the present study is retrospective; unlike prospective randomized clinical studies, we did not construct study designs and estimation for sampling numbers before starting this study.
Line 130: The authors have performed Kaplan-Meier estimator analysis (Figure 6) but did not mention it in the statistical analysis subsection. The authors need to mention this.
Reply: We added the explanation regarding a Kaplan-Meier estimator and the statistical analysis in the Method section. The sentences are as follows “A Kaplan-Meier estimator showing the retreatment-free proportion was constructed and a log-rank test was used to compare a difference of cumulative retreatment-free proportion between 2 groups.”
Results:
Lines 133-134: The authors cite the exact values in the text (68.4 years, 69.7, 75.0, and 77.9) despite being displayed in Table 1. The authors are encouraged to decrease this overlap with the Table and instead summarize these findings. Readers can retrieve this information from the Table.
Reply: We removed the overlapped sentences. Instead, the following sentence “Baseline demographic and clinical characteristics of patients with polypoidal choroidal vasculopathy was shown in Table” was added and summary sentences were also added.
Line 139: Since one Table only exists, authors do not need to write Table 1. Instead, they can add the Table only.
Reply: Thank you for the comment. Table 1 was changed to “Table”.
Line 141: A legend for table 1 is lacking. It is also not enough to put a P-value without mentioning the statistical test that generated this P-value. The authors need to add a legend where they mention what statistical tests were used (chi-squared on genotypes, etc…) and the abbreviation used NA: not applicable.
Reply: We added specific marks to easily understand the applied statistical analysis and explanation of abbreviation below the table.
Minor:
Line 37: Authors must write the acronym for the following genes (ARMS2 and CFH) at their first appearance in the text.
Reply: ARMS2 and CFH were spelled out at first appearance.
Line 55: add a space between consecutive110. Idem in Line 144. Idem in line 139: Genotyping of ARMS2 A69S(rs10490924) and CFH I62V(rs800292).
Reply: Thank you for your pointing, we have added a space in each point.
The quality of figures 4 and 5 is poor. The authors need to handle this manner appropriately.
Reply: Thank you for your pointing. We revised Figure 3-5. We believed that the resolution and legibility of Figures have been improved.
Round 2
Reviewer 1 Report
The manuscript has been improved.
Reviewer 3 Report
The authors answered my comments